# Opal Actigraphy (Activity and Sleep) Measures Compared to ActiGraph: A Validation Study

**DOI:** 10.3390/s23042296

**Published:** 2023-02-18

**Authors:** Vrutangkumar V. Shah, Barbara H. Brumbach, Sean Pearson, Paul Vasilyev, Edward King, Patricia Carlson-Kuhta, Martina Mancini, Fay B. Horak, Kristen Sowalsky, James McNames, Mahmoud El-Gohary

**Affiliations:** 1Department of Neurology, Oregon Health & Science University, Portland, OR 97239, USA; 2APDM Wearable Technologies-a Clario Company, Portland, OR 97201, USA; 3OHSU-PSU School of Public Health, Oregon Health & Science University, Portland, OR 97201, USA; 4Department of Electrical and Computer Engineering, Portland State University, Portland, OR 97207, USA

**Keywords:** activity, sleep, ActiGraph, daily life, validity, wearable sensors, IMUs

## Abstract

Physical activity and sleep monitoring in daily life provide vital information to track health status and physical fitness. The aim of this study was to establish concurrent validity for the new Opal Actigraphy solution in relation to the widely used ActiGraph GT9X for measuring physical activity from accelerometry epic counts (sedentary to vigorous levels) and sleep periods in daily life. Twenty participants (age 56 + 22 years) wore two wearable devices on each wrist for 7 days and nights, recording 3-D accelerations at 30 Hz. Bland–Altman plots and intraclass correlation coefficients (ICCs) assessed validity (agreement) and test–retest reliability between ActiGraph and Opal Actigraphy sleep durations and activity levels, as well as between the two different versions of the ActiGraph. ICCs showed excellent reliability for physical activity measures and moderate-to-excellent reliability for sleep measures between Opal versus Actigraph GT9X and between GT3X versus GT9X. Bland–Altman plots and mean absolute percentage error (MAPE) also show a comparable performance (within 10%) between Opal and ActiGraph and between the two ActiGraph monitors across activity and sleep measures. In conclusion, physical activity and sleep measures using Opal Actigraphy demonstrate performance comparable to that of ActiGraph, supporting concurrent validation. Opal Actigraphy can be used to quantify activity and monitor sleep patterns in research and clinical studies.

## 1. Introduction

Limited physical activity and poor sleep are related to several neurological diseases and result in reduced quality of life [1,2,3,4]. Research has shown a bidirectional relationship between a poor/insufficient sleep and low physical activity levels [5,6,7,8,9]. Hence, accurate assessment of physical activity and sleep is critical for management of many chronic conditions and can help quantify treatment-related changes and track individuals’ overall health status [10]. Traditionally, comprehensive assessments of physical activity and sleep have been conducted in a clinic/laboratory environment that requires expensive equipment and trained staff [11,12,13]. Self-reporting of physical activity and sleep are subjective and depend on individuals’ ability to recall and estimate activity levels and sleep patterns. To overcome these limitations in laboratory and self-reported measures, accelerometer-based wearable technologies have emerged as valid tools to directly and objectively quantify physical activity behaviors and sleep patterns in daily life [14]. Objective assessment of physical activities and sleep patterns with wearable devices shows great promise for application in decentralized clinical trials in patient homes and community settings to capture continuous tracking of daily life health status [15,16,17].

Wearable technology with tri-axial accelerometers has been successfully used to measure physical activity and sleep for decades, and provides a cost-effective and practical solution for continuous tracking of daily life health status [18,19,20]. Clinicians and researchers are often interested in quantifying the time, in minutes, spent in selected levels of physical activity intensity commonly defined as sedentary, light, moderate, vigorous and very vigorous intensity. ActiGraph devices have been among the most commonly used solutions for quantification of physical activities and sleep, with more than 20,000 papers concerning them published to date [21].

Algorithms for computing activity counts in the earlier models of these devices have been published [22,23,24,25,26]. Several activity count algorithms are based on zero-crossing and time above threshold. Freedson et al. established accelerometer count ranges for a physical activity monitor [25]. They provided accelerometer count cut-points for adults that correspond to different activity intensity levels [25]. A decade later, they updated their activity count cut-points to classify physical activity intensity based on tri-axial vector magnitude [24,25]. In addition to physical activity intensity, clinicians and researchers are also interested in quantifying periods of sleep and sleep efficiency in order to distinguish sleep from wakefulness [23].

Several studies demonstrated the validity of accelerometer-based physical activity and sleep measures by comparing them to in-clinic gold-standard assessment [27,28,29,30]. The validity of ActiGraph wrist monitors for the assessment of physical activity and sleep has been extensively studied across many different age groups, genders, and patient populations. Clevenger et al. compared cross-generational ActiGraph devices and assessed their performance in quantifying physical activity [31]. They used GT9X and wGT3X models worn at the hip and on each wrist for 4 days. They reported that epoch-level data from different models were not identical, but most outcomes were strongly related between models and equivalent when reduced to the percentage of time spent in each intensity of activity. They suggested that caution should be exercised when comparing outcome measures among ActiGraph models [31]. Validity varies widely between devices including the Apple Watch, Yamax Digiwalker, iHealth Edge, and Misfit Shine [32]. Valkenet et al. investigated the validity of their accelerometer-based system for measuring physical activity by comparing it with the ActiGraph wGT3X-BT accelerometer [33]. They reported an intraclass correlation coefficient of 0.95, but participants wore the accelerometers for a comparatively short time, that is, one day between 9 am and 4 pm.

Wrist-worn accelerometers have been validated for sleep detection as well. Cole et al. developed and validated automatic scoring methods to distinguish sleep from wakefulness based on wrist activity during overnight polysomnography. They reported that their algorithms correctly distinguished sleep from wakefulness 88% of the time [23]. Neishabouri et al. described the detailed counts algorithms of five generations of ActiGraph devices and published the counts algorithm in ActiGraph’s ActiLife and CentrePoint software [21]. Recently, a sleep detection algorithm based on angular wrist rotation estimated with raw acceleration data was validated using data from multiple accelerometer brands, including Axivity, GENEActiv and ActiGraph [34,35].

Opal^®^ V2 Solutions (APDM Wearable Technologies, a Clario Company) have been used extensively in clinical and academic research to characterize gait, balance, and other aspects of mobility. Opal^®^ V2 wearable sensors contain lightweight inertial measurement units, and can be deployed both in the clinic and remote settings to capture a broad range of digital movement outcomes. Recently, Clario has developed an actigraphy solution to quantify physical activity and sleep measures during daily life. This solution contains one Opal sensor, configured to use the triaxial accelerometer, in a low power mode sampled at 30 hz to collect actigraphy data continuously. The Mobility Lab^®^ software contains algorithms that generate the objective measures of daily activity levels and sleep. In this pilot study, we aimed to validate the Opal Actigraphy measures of physical activity and sleep and assess their agreement with those obtained from two different models of ActiGraph. We chose ActiGraph as a reference because ActiGraph devices (ActiGraph, Pensacola, FL, USA) have been among the most widely used and validated actigraphy devices for over two decades [21]. The main contribution of this study is to provide an alternative solution to characterizing daily free-living physical activity and sleep using Opal, and to demonstrate its concurrent validity compared to the ActiGraph.

## 2. Methods

### 2.1. Participants

Twenty healthy subjects without neurological, musculoskeletal, or sleep disorders participated in the study. The experimental protocol was approved by the Institutional Review Board of the Oregon Health & Science University (eIRB # 15578). All the participants provided informed written consent.

### 2.2. Data Collection

Participants wore 4 wearable devices; 2 on each wrist. One Opal and one ActiGraph GT9X were strapped together and placed on the non-dominant arm, and one ActiGraph GT3X and a second ActiGraph GT9X were strapped together and placed on the dominant arm. Participants were instructed to wear the devices continuously for 7 days (day and night) during their daily activities and sleep. Data collection was performed by team at OHSU, independent of the data analysis team at Clario.

### 2.3. Data Processing

The sensor data were processed by Mobility Lab (Opal) and ActiLife (GT3X and GT9X) to obtain accelerometer epoch counts data and sleep periods (Cole-Kripke algorithm) [18]. The counts were categorized into activity levels (sedentary, light, moderate, vigorous and very vigorous) according to the thresholds established by Freedson et al. with two versions of algorithms, referring to Freedson98 [20] and FreedsonVM3 [19]. We also combined detected sleep periods from each noon–noon period to obtain daily sleep statistics. We chose both algorithms (Freedson98 and FreedsonVM3) as they have been the most widely utilized in the research community and validated for activity and sleep monitoring. They have also been used in by ActiGraph devices and the software used for comparison in our current study.

### 2.4. Statistical Analysis

Concurrent validity is established when a new test or instrument is compared against a previously validated test, often a “gold standard” or that most widely used in the field. In this study, we aim to support validation of Opal Actigraphy by comparing it to the current most widely used wrist-worn sensor in the industry, the ActiGraph. To make the comparison between the devices, we chose several analytic approaches to assess levels of reliability between the Opal and ActiGraph GT9X algorithms. Evidence of high reliability will suggest the concurrent validity of the Opal instrument. Furthermore, we predict that the level of reliability and agreement between the Opal and ActiGraph GT9X will be comparable to the level of agreement between the two ActiGraph devices (GT3X and GT9X), thus providing additional support to validation of the Opal-based actigraphy solution. To establish reliability and agreement, we used intraclass correlation coefficients (ICCs) (two-way mixed effects, absolute agreement) and Bland–Altman plots. Additionally, we calculated mean absolute percentage error (MAPE) to test the performance between Opal versus ActiGraphy GT9X, and ActiGraph GT3X versus Actigraph GT9X. The aim was to test if the difference between the MAPE of Opal vs. GT9X is comparable (within 10% [36,37]) to the difference between MAPE of Opal vs. GT9X for all activity and sleep measures. Bland–Altman and ICC analyses were performed by a non-conflicted biostatistician at OHSU (BB) using STATA 16 software. The figures and all other analyses were produced using R Version 1.1.456 software.

### 2.5. Data Availability Statement

The datasets generated during and/or analyzed during the current study are available from the corresponding author on reasonable request.

## 3. Results

Twenty healthy subjects participated in the study. The mean age of the group was 56 years (standard deviation (SD) = 22 years), with an average height of 65.5 inches (SD = 5.8 inches) and an average weight of 72.6 Kg (SD = 16 Kg). Compliance was high for weekly recordings, with an average of 6.5 days (SD = 0.6 days; min = 6 days and max = 8 days) of recording.

### 3.1. Comparison of Opal and ActiGraph Algorithms for Physical Activity Measures

Freedson98 Algorithm: The ICC values show high reliability and similar range in both comparisons for all activity measures estimated by Freedson98. The ICC ranges across activity levels for Opal versus ActiGraph GT9X were 0.9953–0.9996, and for GT3X versus GT9Xwere 0.9899–0.9999 (see Table 1). Figure 1 shows the Bland–Altman plots comparing Opal and ActiGraph GT9X, and between two versions of ActiGraph (GT3X versus GT9X), side-by-side, for the total activity counts. The degree of agreement between Opal and ActiGraph and between the two ActiGraph models was similar across activity levels (not all plots shown). The biases and limits of agreement for all Freedson98 activity measures are represented by mean and lower limits (LL) and upper limits (UL) in Table 2, respectively. The mean absolute percentage error (MAPE) shows a comparable performance between the Opal and ActiGraph GT9X, and between the two versions of ActiGraph (GT3X versus GT9X) (see Table 2). Specifically, the MAPE for Opal versus ActiGraph GT9X activity measures using the Freedson98 algorithm was within 10% of the MAPE of GT3X versus GT9X.

FreedsonVM3 Algorithm: The ICC values show high reliability and a similar range in both comparisons for all activity measures estimated by FreedsonVM3. The ICC ranges across activity levels for Opal versus ActiGraph GT9X were 0.9916–0.9981, and for GT3X versus GT9X were 0.9957–0.9996 (see Table 1). Figure 2 shows the Bland–Altman plots comparing Opal and ActiGraph GT9X, and between two versions of ActiGraph (GT3X versus GT9X), side-by-side, for the total activity counts. The degree of agreement between Opal and ActiGraph and between the two ActiGraph models was similar across activity levels (not all plots shown). The biases and limits of agreement for all FreedsonVM3 activity measures are represented by mean and lower limits (LL) and upper limits (UL) in Table 3, respectively. The mean absolute percentage error (MAPE) shows a comparable performance between the Opal and ActiGraph GT9X, and between the two versions of ActiGraph (GT3X versus GT9X) (see Table 3). Specifically, the MAPE for Opal versus ActiGraph GT9X activity measures using FreedsonVM3 algorithm was within 10% of the MAPE of GT3X versus GT9X. The mean and SD of all activity measures’ averages across 7 days of all participants are shown in Appendix A.

### 3.2. Comparison of Opal and ActiGraph Algorithms for Sleep Measures

Figure 3 shows a representative plot of sleep detection by Opal (light) versus ActiGraph GT9X (dark) in a single participant over 7 days. ICC values show moderate to high reliability in both comparisons for sleep measures. ICC ranges across sleep measures for Opal versus ActiGraph GT9X: 0.7109–0.9879, and GT3X versus GT9X: 0.7469–0.9001 (see Table 1). Figure 4 shows the Bland–Altman plots comparing Opal and ActiGraph GT9X, and between two versions of ActiGraph (GT3X versus GT9X) side-by-side for all the sleep measures. The degree of agreement between Opal and Actigraph and between the two ActiGraph models was similar across sleep measures. The bias and limits of agreement for all sleep measures are represented by mean and lower limits (LL) and upper limits (UL) in Table 4, respectively. The mean absolute percentage error (MAPE) shows a comparable performance between the Opal and ActiGraph GT9X and between the two versions of ActiGraph (GT3X versus GT9X) (see Table 4). Specifically, the MAPE for Opal versus ActiGraph GT9X sleep measures was within 10% of the MAPE of GT3X versus GT9X, except for the total sleep time. However, in the case of the total sleep time measures, the MAPE for Opal versus GT3X (6%) was lower than that for GT3X versus GT9X (25%). The mean and SD of all sleep measures’ averages across 7 days of all participants are shown in Appendix A.

## 4. Discussion

Opal Actigraphy and ActiLife solution (ActiGraph GT3X and GT9X) generated activity counts, activity intensity levels, and sleep measures during daily life. Comparisons between Opal and ActiGraph GT9X and between ActiGraph GT3X and GT39 were used to assess levels of reliability and agreement. Collectively, in this pilot study, our results indicate high reliability and similar agreement between the Opal and ActiGraph models, and between the two ActiGraph models, lending support for establishing concurrent validity.

*Activity measures:* Total activity counts were slightly underestimated by Opal, and overestimated by GT3X compared to GT9X. This is consistent with the findings by John et al. who observed an overestimation of counts by GT3X compared to GT9X on a treadmill protocol [38]. In contrast, Hwang et al. found that total counts were overestimated by GT9X compared to GT3X [39].

*Sleep measures:* GT3X has been shown to be a reliable and valid to measure sleep compared to a gold standard in-lab, i.e., polysomnography (PSG) [27,30,40]. Sleep efficiency has been shown to be somewhat overestimated by GT3X compared to PSG [27,30,40]. In our results, we also found that sleep efficiency is overestimated by GT3X, and slightly underestimated by Opal compared to GT9X. Similarly, the wake-after-sleep onset time (WASO) has been shown to be underestimated by GT3X compared to PSG [27,30,40]. In our results, we also found that WASO is underestimated by GT3X, and overestimated by Opal compared to GT9X. In contrast, total sleep time is overestimated by GT3X (compared to PSG [27]) and Opal compared to GT9X.

We observed that activity measures yielded higher reliability metrics from ICCs than sleep measures. Further, we observed that for some of the activity and sleep measures, Opal is underestimating while GT3X is overestimating compared to GT9X, and vice versa, with a notable outlier from the Bland–Altman plots. Both of these observed differences could be attributed to the different hardware, sensor characteristics (see Appendix A), and algorithms used by each system. Furthermore, some participants clearly observed the difference in weight (GT3X was bulkier) on one hand, compared to another hand, which might have influenced the measures extracted from the dominant (GT3X versus GT9X) and non-dominant (Opal versus GT9X) hands. Lastly, although the agreement between Opal and ActiGraph GT9X is similar to the agreement between the two versions of ActiGraph, we recommend caution when interpretating these results; for some activity and sleep measures, there appears to be a systematic bias.

There are several limitations of the current study. First, we only assessed the reliability and agreement of activity and sleep measures of the Opal compared to ActiGraph in healthy middle-aged adults, and findings may not be applicable to other cohorts. Second, we only validated Opal’s activity and sleep measures with a small sample of subjects, therefore limiting the strength of the conclusions that can be drawn regarding agreement, especially from the Bland–Altman plots. Finally, we need to investigate the activity and sleep measures for the minimal clinically important difference in order to establish whether the difference between the Opal and GT9X is clinically meaningful. Hence, future studies are needed to strengthen the validity of Opal’s activity and sleep measures on larger, more diverse populations, including those with neurological disorders.

## 5. Conclusions

Our pilot study provided evidence for establishing concurrent validity of t Opal Actigraphy measures of sleep and physical activity with 3D accelerometer data during daily life, as compared to the most widely used ActiGraph. Historically, Opal multi-sensor solutions have enabled high resolution data capture of precise aspects of human mobility via prescribed movement tasks in controlled environments to assess the efficacy and safety of therapeutic interventions. The new and validated Opal Actigraphy single-sensor configuration can provide further insight on how movement impairments captured in clinic translate into real-world quality of life by quantifying overall physical activity levels and sleep durations. Future studies should evaluate Clario’s Opal Actigraphy solution in larger, more diverse cohorts and determine the clinically meaningful important difference in activity and sleep metrics in response to interventions.

## Figures and Tables

**Figure 1 sensors-23-02296-f001:**
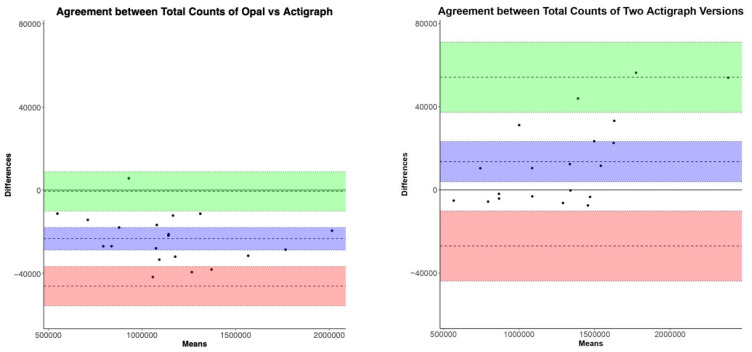
Bland–Altman plots of total counts using Freedson98 algorithm.

**Figure 2 sensors-23-02296-f002:**
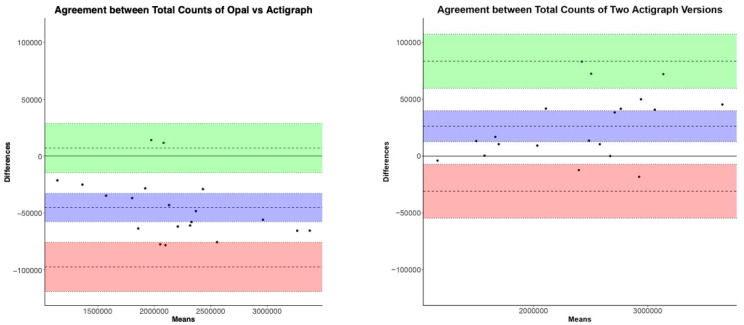
Bland–Altman plots of total counts using FreedsonVM3 algorithm.

**Figure 3 sensors-23-02296-f003:**
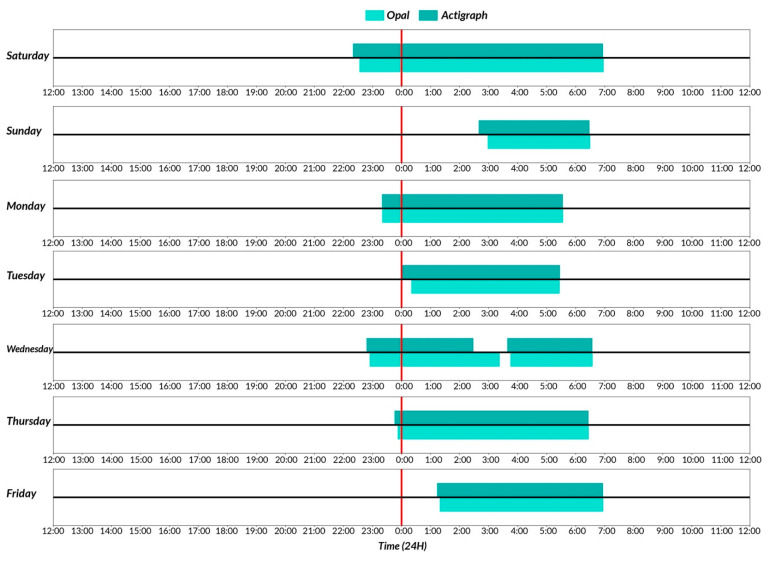
Sleep detection by Opal (light) versus Actigraph GT9X (dark) in a single participant over 7 days.

**Figure 4 sensors-23-02296-f004:**
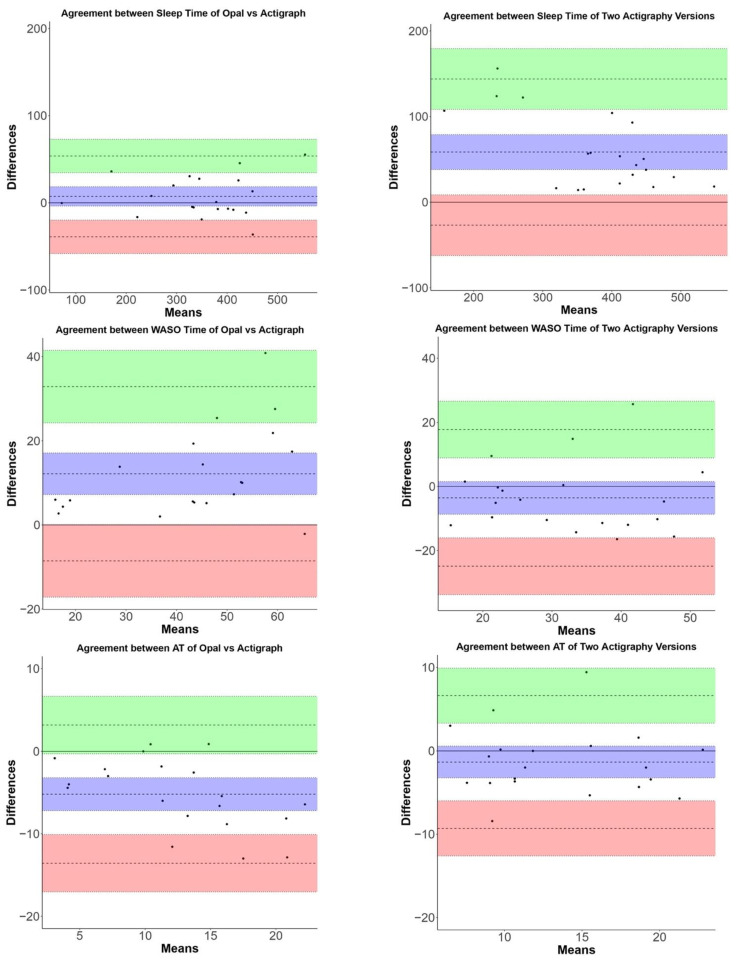
Bland–Altman plots of sleep measures.

**Table 1 sensors-23-02296-t001:** ICC (two-way mixed effects, absolute agreement) for activity and sleep measures.

Metric	Opal versus ActiGraph GT9X ICC [95% CI]	Actigraph GT3X versus ActiGraph GT9X ICC [95% CI]
Activity measures (Freedson 98)		
Total counts (#)	0.9986 [0.8683–0.9997]	0.9992 [0.9968–0.9997]
Light activity (min)	0.9977 [0.9931–0.9991]	0.9993 [0.9981–0.9997]
Moderate activity (min)	0.9977 [0.9344–0.9995]	0.9997 [0.9993–0.9999]
Vigorous activity (min)	0.9954 [0.9807–0.9985]	0.9899 [0.9748–0.9960]
Very vigorous activity (min)	0.9996 [0.9990–0.9999]	0.9999 [0.9996–0.9999]
Sedentary activity (min)	0.9991 [0.9920–0.9997]	0.9995 [0.9935–0.9999]
Activity measures (Freedson VM3)		
Total counts (#)	0.9978 [0.8813–0.9995]	0.9991 [0.9932–0.9997]
Light activity (min)	0.9981 [0.9324–0.9996]	0.9996 [0.9983–0.9998]
Moderate activity (min)	0.9968 [0.9891–0.9984]	0.9990 [0.9974–0.9996]
Vigorous activity (min)	0.9971 [0.9927–0.9988]	0.9976 [0.9901–0.9992]
Very vigorous activity (min)	0.9916 [0.9510–0.9974]	0.9957 [0.9845–0.9985]
Sleep measures		
Total sleep time (min)	0.9879 [0.9696–0.9952]	0.8703 [−0.0178–0.9665]
Wake after sleep onset (min)	0.7813 [−0.0530–0.9349]	0.7469 [0.3813–0.8985]
Awakenings (#)	0.7109 [−0.1614–0.9109]	0.8272 [0.5733–0.9310]
Avg. awakening length (min)	0.8480 [0.6179–0.9397]	0.9001 [0.7518–0.9602]
Sleep efficiency (%)	0.8786 [0.1311–0.9668]	0.8436 [−0.1312–0.9608]
Total counts (#)	0.8948 [0.6800–0.9612]	0.7916 [0.4723–0.9176]

**Table 2 sensors-23-02296-t002:** Performance between Opal and Actigraph GT9X, and between Actigraph GT3X and GT9X on activity measures (FreedsonVM3).

	Opal vs. ActiGraph GT9X					ActiGraph GT3X vs. ActiGraph GT9X				
Metric	Mean	SD	LB	UB	MAPE	Mean	SD	LB	UB	MAPE
Total counts (#)	−23,391.81	11,636.71	−46,199.77	−583.85	2.15	13,620.79	20,727.51	−27,005.13	54,246.72	1.24
Light activity(min)	2.78	5.34	−7.68	13.24	0.95	1.67	3.72	−5.62	8.97	0.59
Moderate activity (min)	−5.36	3.86	−12.93	2.21	2.63	0.66	2.71	−4.66	5.97	1.11
Vigorous activity (min)	−0.61	0.89	−2.37	1.14	20.83	0.57	2.05	−3.44	4.58	22.34
Very vigorous activity (min)	−0.11	0.25	−0.60	0.38	29.38	0.07	0.16	−0.24	0.38	35.18
Sedentary activity (min)	3.30	3.32	−3.21	9.81	0.52	−2.96	2.65	−8.16	2.23	0.45

**Table 3 sensors-23-02296-t003:** Performance between Opal and Actigraph GT9X, and between Actigraph GT3X and GT9X on activity measures (FreedsonVM3).

	Opal vs. ActiGraph GT9X					ActiGraph GT3X vs. ActiGraph GT9X				
Metric	Mean	SD	LB	UB	MAPE	Mean	SD	LB	UB	MAPE
Total counts (#)	−45,271.40	26,682.87	−97,569.83	7027.02	2.16	26,295.33	29,198.24	−30,933.22	83,523.88	1.17
Light activity(min)	5.90	3.96	−1.86	13.66	0.57	−2.17	3.41	−8.85	4.52	0.26
Moderate activity (min)	−3.42	5.91	−15.00	8.16	1.95	−1.30	3.90	−8.95	6.36	1.27
Vigorous activity (min)	−1.15	4.07	−9.12	6.83	6.54	2.60	3.82	−4.89	10.10	4.25
Very vigorous activity (min)	−1.33	1.62	−4.50	1.84	25.73	0.86	1.38	−1.84	3.56	16.52

**Table 4 sensors-23-02296-t004:** Performance of Opal versus Actigraph GT9X and Actigraph GT3X versus GT9X on sleep measures. Abbreviations: WASO = wake after sleep onset; AT = awakening time; AL = awakening length; SE = sleep efficiency.

	Opal vs. ActiGraph GT9X					ActiGraph GT3X vs. GT9X				
Metric	Mean	SD	LB	UB	MAPE	Mean	SD	LB	UB	MAPE
Total sleep time (min)	7.32	23.64	−39.02	53.66	5.74	58.51	43.57	−26.89	143.91	25.32
Wake after sleep (min)	12.15	10.57	−8.58	32.87	35.60	−3.59	10.90	−24.96	17.78	28.59
Awakenings (#)	−5.19	4.27	−13.56	3.18	34.29	−1.34	4.06	−9.30	6.62	27.64
Avg. awakening (min)	0.09	0.61	−1.11	1.29	17.40	−0.09	0.39	−0.85	0.68	10.79
Sleep efficiency (%)	−2.68	2.29	−7.16	1.80	3.06	2.48	1.63	−0.72	5.68	2.95
Counts (#)	2321.51	3976.71	−5472.83	10,115.86	18.94	−600.24	4537.03	−9492.83	8292.34	24.56

## Data Availability

Data from this study will be made available upon reasonable request to the corresponding author.

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
