# Peer review of "Opal Actigraphy (Activity and Sleep) Measures Compared to ActiGraph: A Validation Study"

_sensors, 2023, doi:10.3390/s23042296_

Round 1

Reviewer 1 Report

The research work aims to establish concurrent validity for the Opal Actigraphy solution with the traditional gold standard ActiGraph for measuring physical activity and sleep in daily life. Although it has some merits, there are some key comments to be addressed by authors. Therefore, a major revision is recommended.

Comment 1. Refer to the journal’s template. Update the format of the paper, for examples, list of authors, spacing, in-text citations, captions of the subfigures, remarks after conclusion.
Comment 2. Abstract:
(a) Refer to the journal’s template, abstract should be with
a single paragraph of about 200 words maximum.
(b) Emphasize the research implications.
Comment 3. Keywords, more terms should be included to better reflect the scopes of the paper.
Comment 4. Section 1 Introduction:
(a) Elaborate the relationship between physical activity and sleep.
(b) Enhance the literature review with a detailed discussion on the methodology, results, and limitations of the existing works.
(c) Summarize the research contributions.
Comment 5. Section 2 Methods:
(a) Elaborate the data collection process if the dataset was managed by authors’ research team.
(b) Elaborate the methodology for the concurrent validity.
Comment 6. Section 3 Results:
(a) Justify the selections of Freedson98 Algorithm, and FreedsonVM3 Algorithm.
(b) The supplementary figures and tables should be moved to main-text because the contents are very important.
Comment 7. Section 4 Discussion:
(a) The contents should be supported by references.
(b) Comparison between authors’ research work and existing works is expected.
Comment 8. Elaborate future research directions.

Author Response

Please find the rebuttal document attached.

Reviewer 2 Report

Wearable device comparisons add to the body of knowledge related to progress in product development of devices that can facilitate clinical decision-making, and the current work clearly takes a step in that direction by comparing two commercially available devices.

The discussion of the ActiGraph as a gold standard for sleep is perplexing since it implies this is a clinical gold and it is unclear that sleep lab polysomnography no longer holds that claim.  Indeed, several studies in the literature show that actigraphy is “comparable” but leaves room for improvement.  In this way, the popularity of sales might make for a non-clinical gold standard, but not for a clinical gold standard.  The manuscript has the potential to be misleading in this area. The credibility of the paper would be improved by care with the phrase "gold standard."

Lines 99-103 alerted the reader that the authors have potential conflict of interest - though it is declared --and so possibly re-phrasing this section for a more modest factual statement would make this paper better aligned with a peer-reviewed article (rather than what would be in a trade journal).  The bias is subtle, and it is the authors’ prerogative to claim if data can be added.

Some opinions are that wearble device performance can be fairly optimistic, and so for the paper to have greater credibility with readership it may help to suggest this is a “pilot study” since there are only 20 subjects.  To claim device “validation” with 20 subjects would be inconsistent with other many other wearble studies with hundreds if not thousands of subjects.  This tends to fall within “burden of proof” in authorship when claiming a new device is comparable to a gold standard.

Comparison of results is frequently described as “similar” and it would help to be more specific.  Table 1 is confusing on the intent as well as the format is very confusing -- there are so may parameters and they are not prioritized as to what is the critical measure the authors’ feel is most important. Is would help the reader to not assume outcomes are self-evident from a large matrix of numbers.  Further, if you have a gold standard that has variation (GT3X vs GT9X), are you suggesting that if the difference between Opal vs GT9X is no worse that the variance in the gold standard – this is unclear. The narrative for the figures is not descriptive: terms like “slightly lower” and “similar” are in contrast (high level) to all the data that has evidently been collected and statistically analyzed.  Based on the literature, what specific numerical differences would be expected in say, the Bland-Altman plots or MAE for the results to be considered similar?

It is unclear if all experimental data is available for third party verification and reproducibility tests.  This would be an essential element to a research work that is centered on a validation study.

Device technology differences are not adequately described. We are aware that for a given phenomenon, that, for instance, mean absolute error is a measure of errors between paired observations. But the reader might have greater insight on the significance of MAE if the reason for measurements were more evident or even suggested.  As is, the “black box” approach is not as compelling in the clinical research arena, and even in light of the manuscript conclusion that the device is a commercially acceptable alternative some commentary on where measurement techniques differ would be very helpful.

Author Response

(The authors gave the same response as above.)

Round 2

Reviewer 1 Report

I have some minor follow-up comments.
Follow-up comment 1: Regarding Abstract, ensure the terms are precise to reflect the scopes of the paper. For examples, activity, sleep, and validity are too general.
Follow-up comment 2: Table 1, proper spacing is needed for "Light activity(min)" and "Light activity(min)"
Follow-up comment 3: Conclusion is too short. Please elaborate with the emphasis of research contributions, research implications, and future research directions.

Reviewer 2 Report

The revised paper addresses concerns raised previously.
